# Leveraging single-cell ATAC-seq and RNA-seq to identify disease-critical fetal and adult brain cell types

**Samuel S. Kim** [1,2] ✉, **Buu Truong** [2,3,8] ✉, **Karthik Jagadeesh**[2,8], **Kushal K. Dey** [2,4,8], **Amber Z. Shen**[5], **Soumya Raychaudhuri** [6], **Manolis Kellis** [1] & **Alkes L. Price** [1,2,3,7] ✉

Prioritizing disease-critical cell types by integrating genome-wide association studies (GWAS) with functional data is a fundamental goal. Single-cell chromatin accessibility (scATAC-seq) and gene expression (scRNA-seq) have characterized cell types at high resolution, and studies integrating GWAS with scRNA-seq have shown promise, but studies integrating GWAS with scATAC-seq have been limited. Here, we identify disease-critical fetal and adult brain cell types by integrating GWAS summary statistics from 28 brain-related diseases/traits (average $N = 298$ K) with 3.2 million scATAC-seq and scRNA-seq profiles from 83 cell types. We identified disease-critical fetal (respectively adult) brain cell types for 22 (respectively 23) of 28 traits using scATAC-seq, and for 8 (respectively 17) of 28 traits using scRNA-seq. Significant scATAC-seq enrichments included fetal photoreceptor cells for major depressive disorder, fetal ganglion cells for BMI, fetal astrocytes for ADHD, and adult VGLUT2 excitatory neurons for schizophrenia. Our findings improve our understanding of brain-related diseases/traits and inform future analyses.

Genome-wide association studies (GWAS) have been successful in identifying disease-associated loci, occasionally producing valuable functional insights[1,2]. Identifying disease-critical cell types (defined as cell types whose biology critically influences the etiology of disease) is a fundamental goal for understanding disease mechanisms, designing functional follow-ups, and developing disease therapeutics[3]. Several studies have identified disease-critical tissues and cell types using bulk chromatin[4–9] and/or gene expression data[8,10–12]. With the emergence of single-cell profiling of diverse tissues and cell types[13–17], several studies have integrated GWAS data with single-cell chromatin accessibility (scATAC-seq)[16–20] and single-cell gene expression (scRNA-seq)[10,21,22].

However, compared to scRNA-seq data, scATAC-seq data has been less well-studied for identifying disease-critical cell types. In addition, while it is widely known that biological processes in the human brain vary with developmental stage[23–27], the impact on disease risk of cell types in different developmental stages of the brain has not been widely explored. This motivates further investigation of scATAC-seq and scRNA-seq data at different developmental stages.

Here, we infer disease-critical cell types by analyzing scATAC-seq and scRNA-seq data derived from single-cell profiling of over 3 million cells from fetal and adult human brains. We analyze 83 brain cell types from 4 single-cell datasets[14–17] across 28 brain-related diseases and

[1]Department of Electrical Engineering and Computer Science, Massachusetts Institute of Technology, Cambridge, MA, UK. [2]Department of Epidemiology, Harvard T.H. Chan School of Public Health, Boston, MA, UK. [3]Program in Medical and Population Genetics, Broad Institute of MIT and Harvard, Cambridge, MA, UK. [4]Computational and Systems Biology Program, Sloan Kettering Institute, Memorial Sloan Kettering Cancer Center, New York, NY, USA. [5]Department of Mathematics, Massachusetts Institute of Technology, Cambridge, MA, USA. [6]Division of Genetics, Department of Medicine, Brigham and Women's Hospital and Harvard Medical School, Boston, MA, USA. [7]Department of Biostatistics, Harvard T.H. Chan School of Public Health, Boston, MA, USA. [8]These authors contributed equally: Buu Truong, Karthik Jagadeesh, Kushal K. Dey. ✉e-mail: samuelkim484@gmail.com; btruong@broadinstitute.org; btruong@hsph.harvard.edu; aprice@hsph.harvard.edu

complex traits (average $N = 298$ K). We determine that both scATAC-seq and scRNA-seq data are highly informative for identifying disease-critical cell types; surprisingly, scATAC-seq data is somewhat more informative in the data that we analyze.

## Results

### Overview of methods

We define a *cell-type annotation* as an assignment of a binary or probabilistic value between 0 and 1 to each SNP in the 1000 Genomes European reference panel[28], representing the estimated contribution of that SNP to gene regulation in a particular cell type. Here, we constructed cell-type annotations for 4 datasets: (1) fetal brain scATAC-seq[16] (number of cell types ($C$) = 14), (2) fetal brain scRNA-seq data[15] ($C = 34$), (3) adult brain scATAC-seq[17] ($C = 18$), and (4) adult brain scRNA-seq data[14] ($C = 17$) (see Web resources).

For scATAC-seq cell-type annotations, we used the chromatin accessible peaks (MACS2[29] peak regions) provided by refs. 16,17. These peaks correspond to accessible regions for transcription factor binding, indicative of active gene regulation. For scRNA-seq cell-type annotations, we used the sc-linker pipeline[22] to construct probability scores annotating SNPs linked to specifically expressed genes in a given cell type[8] (compared to other brain cell types) using brain-specific enhancer-gene links[7,22,30,31].

We assessed the heritability enrichments of the resulting cell-type annotations by applying S-LDSC[11] across 28 distinct brain-related diseases and traits (pairwise genetic correlation <0.9; average $N = 298$ K; Supplementary Data 1) to identify significant disease-cell type associations (Fig. 1). For each disease-cell type pair, we estimated the heritability enrichment[11] (the proportion of heritability explained divided by the annotation size, which is defined as the average annotation value for probabilistic annotations) and standardized effect size[32] ($\tau^*$, defined as the proportionate change in per-SNP heritability associated to a one standard deviation increase in the value of the annotation, conditional on other annotation). We assessed the statistical significance of disease-cell type associations based on per-dataset FDR < 5% (for each of 4 datasets, aggregating diseases, and cell types) based on p-values for positive $\tau^*$, as $\tau^*$ quantifies effects that are unique to the cell-type annotation. We conditioned the analyses on a broad set of coding, conserved, and regulatory annotations from the baseline model[11] (Supplementary Data 3). For scATAC-seq annotations, we additionally conditioned on the union of open chromatin regions across all brain cell types in each data set analyzed (consistent with recent unpublished work[33,34], but different from[17,19]), a conservative step to ensure cell-type specificity (see Discussion). For scRNA-seq annotations, we additionally conditioned on the union of brain-specific enhancer-gene links across all genes analyzed (consistent with[21]).

We did not condition on the LD-related annotations included in the baseline-LD model of refs. 32,35, as these annotations reflect the action of negative selection, which may obscure cell-type-specific signals[36]. Further details are provided in the Methods section. We have publicly released all celltype annotations analyzed in this study and source code for all primary analyses (see Data and code availability).

### Identifying disease-critical cell types using fetal brain data

We sought to identify disease-critical cell types using fetal brain data, across 28 distinct brain-related diseases and traits (Supplementary Data 1). We analyzed 14 fetal brain cell types from scATAC-seq data[16] (donor size = 26; fetal age of 72-129 days) and 34 fetal brain cell types from scRNA-seq data[15] (donor size = 28; fetal age of 89-125 days) (Supplementary Data 4; see Methods).

We first analyzed fetal brain scATAC-seq data spanning 14 cell types[16]. We identified 152 significant disease-cell type pairs (FDR < 5% for positive $\tau^*$ conditional on other annotations; Table 1, Table 2, Fig. 2A, Supplementary Data 5). Consistent with previous genetic studies[8,17,21], we identified strong enrichments of excitatory (i.e., glutamatergic) neurons in psychiatric and neurological disorders, including schizophrenia (SCZ), major depressive disorder (MDD), and attention deficit hyperactivity disorder (ADHD) (Fig. 2A); in particular, the role of glutamatergic neurons in MDD is well-supported, as evident from decreased glutamatergic neurometabolite levels in subjects with depression[37]. Consistent with[19], we also identified enrichment of inhibitory (GABAergic) neurons in SCZ; this result is supported by GABA dysfunction in the cortex of schizophrenia cases[38].

Our results also highlight several disease-cell type associations that have not (to our knowledge) previously been reported in analyses of genetic data (Table 2). First, photoreceptor cells were enriched in insomnia. Photoreceptor cells, present in the retina, convert light into signals to the brain, and thus play an essential role in circadian rhythms[39], explaining their potential role in insomnia. Second, photoreceptor cells were also enriched in MDD, a genetically uncorrelated trait ($r = -0.01$ with insomnia) (as well as neuroticism; $r = 0.68$ with MDD). Recent studies support the relationship between the degeneration of photoreceptors and anxiety and depression[40]. Third, ganglion cells were enriched in BMI. Ganglion cells are the projection neurons of the retina, relaying information from bipolar and amacrine cells to the brain. Patients with morbid obesity display significant differences in retinal ganglion cells, retinal nerve fiber layer thickness, and choroidal thickness[41]. Fourth, purkinje neurons were enriched in

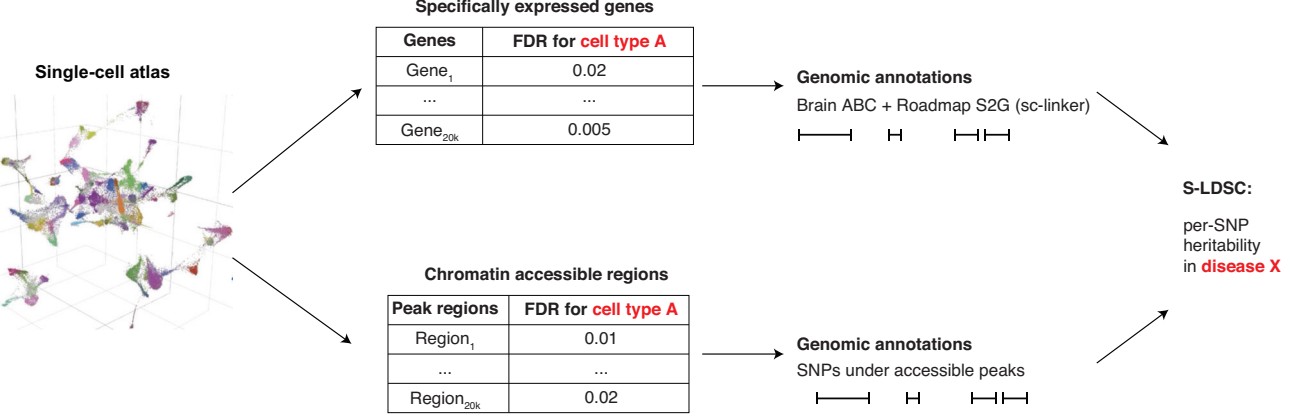

**Fig. 1 | Overview of methods and analyses.** We describe the overview of methods building cell-type annotations from single-cell sequencing datasets (UMAP from[16]) and evaluating disease informativeness applying S-LDSC across GWAS summary statistics. ABC+Roadmap S2G refers to the brain-specific SNPsto-Genes linking strategy using enhancer-gene links[7,21,29,30]. We separately analyzed fetal and adult brain data.

**Table 1 | Summary of findings**

|  | Fetal scATAC | Fetal scRNA | Adult scATAC | Adult scRNA |
|---|---|---|---|---|
| **Brain cell types** | 14 | 34 | 18 | 17 |
| **Total disease-cell type pairs** | 392 | 952 | 504 | 476 |
| **Significant disease-cell-type pairs** | 152 | 9 | 168 | 64 |
| **Significant diseases (out of 28)** | 22 | 8 | 23 | 17 |
| **Data source** | ref. 16 | ref. 15 | ref. 17 | ref. 14 |

For each of 4 single-cell chromatin and gene expression data sets analyzed, we report the number of brain cell types analyzed, the total number of disease-cell type pairs analyzed (based on 28 diseases/traits), the number of significant disease-cell type pairs (FDR < 5% for positive $\tau^*$), and the number of diseases/traits with a significant disease-cell type pair.

**Table 2 | Notable disease-cell type associations**

| Disease/trait | Cell type | Data source | $\tau^*$ (SE) | p-value($\tau^*$) | q-value |
|---|---|---|---|---|---|
| Insomnia[75] | Photoreceptor cells | Fetal brain scATAC | 0.81 (0.23) | $4.58 \times 10^{-4}$ | $2.02 \times 10^{-3}$ |
| MDD[76] | Photoreceptor cells | Fetal brain scATAC | 0.67 (0.17) | $8.45 \times 10^{-5}$ | $5.26 \times 10^{-4}$ |
| SCZ[77] | Inhibitory neurons | Fetal brain scATAC | 0.98 (0.22) | $6.14 \times 10^{-6}$ | $7.08 \times 10^{-5}$ |
| BMI[76] | Ganglion cells | Fetal brain scATAC | 0.55 (0.09) | $8.72 \times 10^{-10}$ | $6.84 \times 10^{-8}$ |
| Insomnia[78] | Purkinje neurons | Fetal brain scATAC | 0.73 (0.21) | $6.01 \times 10^{-4}$ | $2.48 \times 10^{-3}$ |
| ADHD[79] | Astrocytes | Fetal brain scATAC | 1.05 (0.32) | $9.68 \times 10^{-4}$ | $3.72 \times 10^{-3}$ |
| Reaction time[45] | Ganglion cells | Fetal brain scRNA | 0.45 (0.14) | $1.26 \times 10^{-3}$ | $3.93 \times 10^{-2}$ |
| MDD[80] | BDNF excitatory neurons | Adult brain scATAC | 1.31 (0.20) | $1.14 \times 10^{-10}$ | $4.10 \times 10^{-9}$ |
| Bipolar disorder[81] | Parvalbumin interneurons | Adult brain scATAC | 1.23 (0.28) | $7.35 \times 10^{-6}$ | $8.23 \times 10^{-5}$ |
| SCZ[77] | VGLUT2 excitatory neurons | Adult brain scATAC | 1.31 (0.24) | $4.14 \times 10^{-8}$ | $7.45 \times 10^{-7}$ |
| Intelligence[82] | Corticofugal projection neurons | Adult brain scRNA | 0.76 (0.14) | $1.17 \times 10^{-7}$ | $1.39 \times 10^{-5}$ |

We report the disease/trait, cell type, data source, standardized effect size ($\tau^*$), p-value for positive $\tau^*$, and FDR q-value for selected results. Full results are reported in Supplementary Data 5, Supplementary Data 7, Supplementary Data 15, Supplementary Data 16. A description of diseases/traits analyzed is provided in Supplementary Data 1. *MDD* major depressive disorder, *SCZ* schizophrenia, *BMI* body mass index, *ADHD* attention deficit hyperactivity disorder.

insomnia (as well as sleep duration ($r = -0.03$ with insomnia) and chronotype ($r = -0.03$ with insomnia; $r = -0.01$ with sleep duration)). While purkinje neurons play a major role in controlling motor movement, they also regulate the rhythmicity of neurons, consistent with a role in impacting sleep[42]. Fifth, astrocytes were enriched in ADHD. Astrocytes perform various functions including synaptic support, control of blood flow, and axon guidance[43]. In particular[44], highlighted the role of the astrocyte Gi-coupled $GABA_B$ pathway activation resulting in ADHD-like behaviors in mice.

We next analyzed fetal brain scRNA-seq data spanning 34 cell types[15] (of which 13 were also included in fetal brain scATAC-seq data; Supplementary Data 6). We identified 9 significant disease-cell type pairs (FDR < 5% for positive $\tau^*$ conditional on other annotations; Table 1, Table 2, Fig. 2B, Supplementary Data 7). When restricting to the 7 significant disease-cell type pairs corresponding to the 13 cell types included in both scATAC-seq and scRNA-seq data, 6 of 7 were also significant in analyses of scATAC-seq data. In particular, the enrichment of retinal ganglion cells in reaction time ($p = 1.26 \times 10^{-3}$ in scRNA-seq data, FDR $q = 0.039$) was non-significant in scATAC-seq data ($p = 0.028$, FDR $q = 0.060$). The enrichment of retinal ganglion cells in reaction time has not (to our knowledge) previously been reported in analyses of genetic data. Previous genetic analyses have focused on enrichments of cerebellum and brain cortex in reaction time[45], but the involvement of retinal ganglion cells in receiving visual information and propagating it to the rest of the brain is consistent with a role in visual reaction time[46].

We compared the results for 13 fetal brain cell types included in both the scATAC-seq and scRNA-seq datasets (Fig. 2C and Supplementary Data 8). While scATAC-seq and scRNA-seq cell-type annotations for matched cell types were approximately uncorrelated to each other ($r = 0.01$–$0.06$; Supplementary Data 9), the corresponding $-\log_{10}$(p-values) for positive $\tau^*$ were moderately correlated ($r = 0.24$), confirming the shared biological information. We observed more

significant p-values for scATAC-seq than for scRNA-seq in these data sets (see Discussion).

We performed 5 secondary analyses. First, we analyzed enrichments of both scATAC-seq and scRNA-seq brain cell types in 6 control (non-brain-related) diseases and complex traits. As expected, we did not identify any significant enrichments (Supplementary Data 10 and Supplementary Data 11). Furthermore, Q-Q plots confirmed a null distribution of P-values for nonzero $\tau^*$ (Figure S1), validating the normality assumption of $\tau^*$ divided by its jackknife standard error. Second, we performed gene set enrichment analysis using GREAT[47] for both scATAC-seq and scRNA-seq cell-type annotations. As expected, we identified significant enrichments in relevant gene sets (e.g.,"photoreceptor cell differentiation" for photoreceptor cells from scATAC-seq; "negative regulation of cell projection organization" for ganglion cells from scRNA-seq; Supplementary Data 12). Third, for the fetal scRNA-seq data[15], we constructed annotations based on a ±100 kb window-based strategy (previously used in ref. 8) instead of brain-specific enhancer-gene links[7,30,31] (used in ref. 22). We identified 22 significant disease-cell type pairs (Supplementary Data 13), vs. only 9 using brain-specific enhancergene links (although we observed a much stronger opposite trend in adult scRNA-seq data; see below). Fourth, we analyzed bulk chromatin data (7 chromatin marks) spanning 5 fetal brain tissues[9] (age 52–142 days). We identified 541 significant disease-tissue-chromatin mark triplets spanning 26 of 28 brain-related traits (Supplementary Data 14). These results are included for completeness, but cannot achieve the same cell-type specificity as analyses of single-cell data. Fifth, we modified our analyses of scRNA-seq data by constructing binary annotations by converting all positive probability scores to 1. We determined that this produced results that were similar to but slightly worse than our primary analysis involving probability scores ($\tau^*$ regression slope = 0.677) (Figure S2). Interestingly, most nonzero probability scores are either close to 0 or close to 1 (Figure S3); the fact that binarizing the probability scores

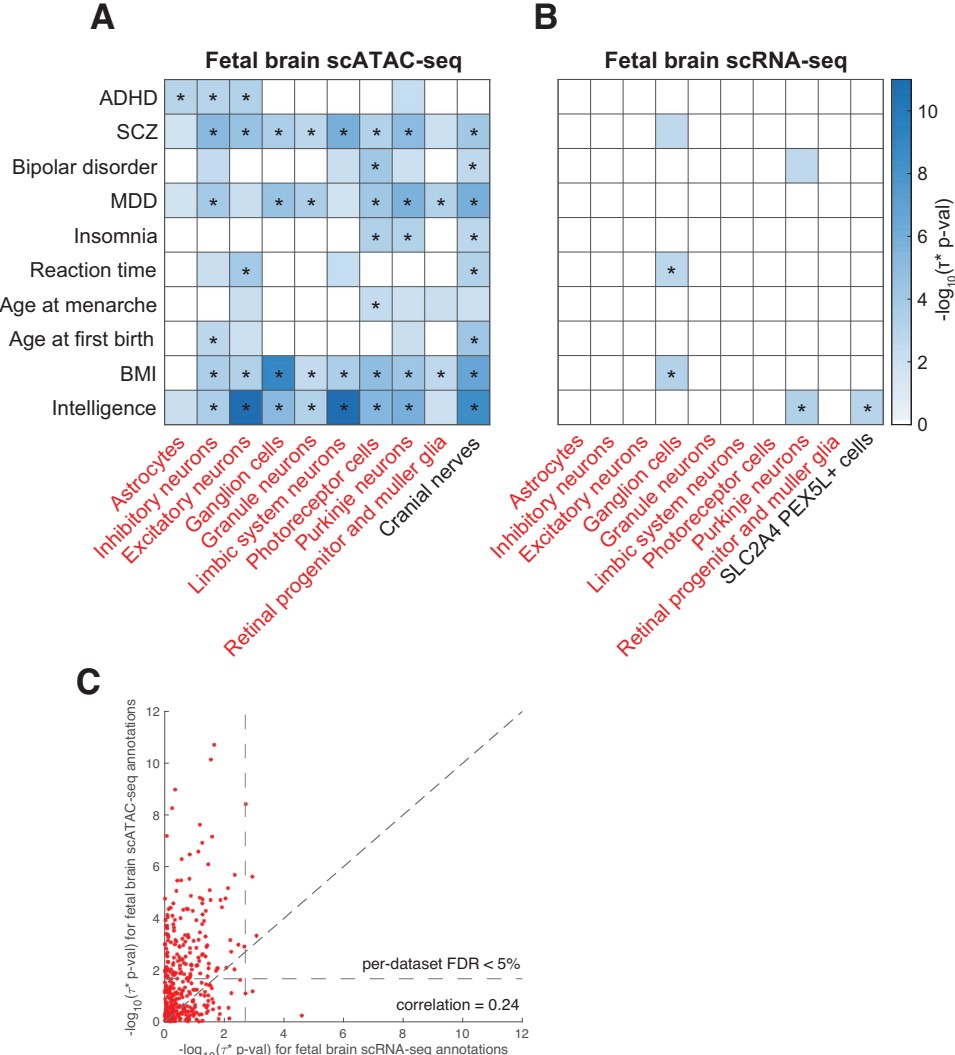

**Fig. 2 | Disease enrichments of cell-type annotations derived from fetal brain.** We report **A** −log$_{10}$ p-values for positive $\tau^*$ for a subset of 10 (of 28) diseases/traits and 10 (of 14) fetal brain scATAC-seq cell type annotations; **B** −log$_{10}$ p-values for positive $\tau^*$ for a subset of 10 (of 28) diseases/traits and 10 (of 34) fetal brain scRNA-seq cell type annotations; and **C** comparison of results for 13 cell types included in both fetal brain scATAC-seq and scRNA-seq data. In **A**, **B**, only statistically significant results (FDR > 5%) are colored ( − log$_{10}$(p-value) ≥ 1.67 for scATAC-seq, ≥ 2.70 for scRNA-seq). In **A**, **B**, cell types appearing in both datasets are denoted in red font. Numerical results for all diseases/traits and cell types are reported in Supplementary Data 5, Supplementary Data 7, and Supplementary Data 8. * denotes Bonferroni-significant results. ADHD attention deficit hyperactivity disorder, SCZ schizophrenia, MDD major depressive disorder, BMI body mass index.

produces slightly worse results implies that nonzero probability scores that are close to 0 are less informative than nonzero probability scores that are close to 1.

**Identifying disease-critical cell types using adult brain data**
We sought to identify disease-critical cell types using adult brain data, across 28 distinct brain-related diseases and traits (Supplementary Data 1). Analysis of brains with varying developmental stages might elucidate biological mechanisms, as brains undergo changes in cell type composition and gene expression during development[26,27]. We analyzed 18 adult brain cell types from scATAC-seq data[17] (donor size = 10; age 38-95 years) and 17 adult brain cell types from scRNA-seq data[14] (donor size = 31; age 4–22 years) (Supplementary Data 4; see Methods). For brevity, we use the term adult to refer to child and adult donors who have surpassed the fetal development stage.

We first analyzed adult brain scATAC-seq data spanning 18 cell types[17]. We identified 168 significant disease-cell type pairs (FDR < 5% for positive $\tau^*$ conditional on other annotations; Table 1, Table 2, Fig. 3A, Supplementary Data 15). Consistent with previous genetic studies[8,17,19,34],

we identified strong enrichments of excitatory neurons in SCZ and bipolar disorder (genetic correlation $r$ = 0.70) (Fig. 3A). Although an analysis of mouse scATAC-seq identified a significant enrichment of excitatory neurons in SCZ cases vs. bipolar cases[19], we did not replicate this finding ($p$ = 0.66 for positive $\tau^*$; Supplementary Data 15).

Our results also highlight disease-cell type associations that have not (to our knowledge) previously been reported in analyses of genetic data (Table 2). First, brain-derived neurotrophic factor (BDNF) excitatory neurons were highly enriched in MDD (and several other diseases/traits, including bipolar disorder and SCZ). BDNF is involved in supporting survival of existing neurons and differentiating new neurons, and decreased BDNF levels have been observed in untreated MDD[48], bipolar[49] and SCZ cases[50]. Previous studies identified an enrichment of excitatory neurons in MDD[34]. Second, parvalbumin interneurons were enriched in bipolar disorder (and SCZ). Decreased expression and diminished function of parvalbumin interneurons in regulating balance of excitation and inhibition have been observed in bipolar disorder and SCZ cases[51,52]. Third, vesicular glutamate transporter (VLUGT2) excitatory neurons were enriched in SCZ (as well as

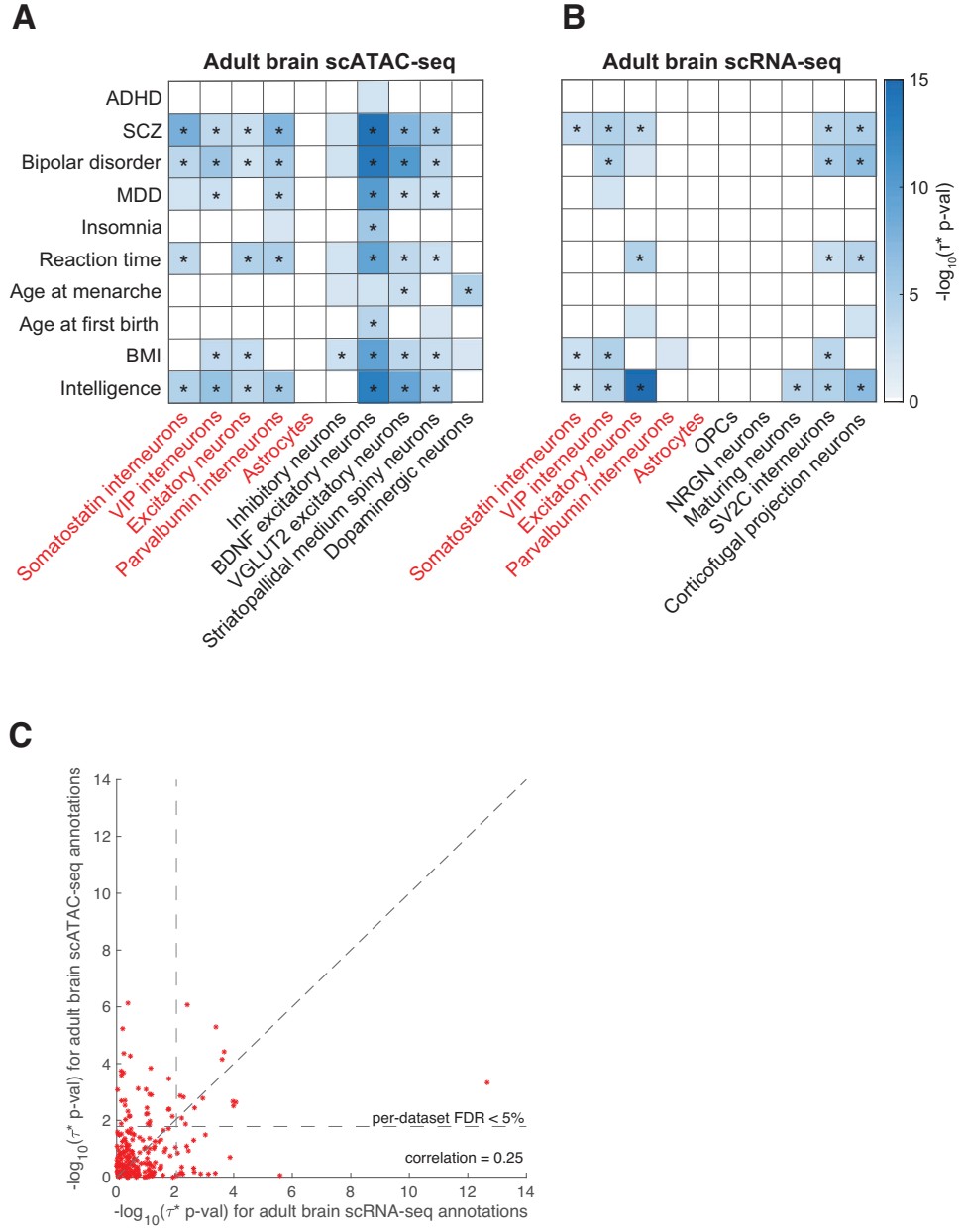

**Fig. 3 | Disease enrichments of cell-type annotations derived from adult brain.** We report **A** $-\log_{10}$ $p$-values for positive $\tau^*$ for a subset of 10 (of 28) diseases/traits and 10 (of 18) adult brain scATAC-seq cell type annotations; **B** $-\log_{10}$ $p$-values for positive $\tau^*$ for a subset of 10 (of 28) diseases/traits and 10 (of 17) adult brain scRNA-seq cell type annotations; **C** comparison of results for 8 cell types included in both adult brain scATAC-seq and scRNA-seq data. In **A**, **B**, only statistically significant results (FDR > 5%) are colored ($-\log_{10}(p$-value) ≥ 1.79 for scATAC-seq, ≥ 2.04 for scRNA-seq). In **A**, **B**, cell types appearing in both datasets are denoted in red font. Numerical results for all diseases/traits and cell types are reported in Supplementary Data 15, Supplementary Data 16, Supplementary Data 17. * denotes Bonferroni-significant results. ADHD attention deficit hyperactivity disorder, SCZ schizophrenia, MDD major depressive disorder, BMI body mass index.

bipolar disorder and intelligence). VLUGT2 knock-out mice display glutamatergic deficiency, diminished maturation of pyramidal neuronal architecture, and impaired spatial learning and memory[53], supporting a role in SCZ and intelligence.

We next analyzed adult brain scRNA-seq data spanning 17 cell types[14] (of which 8 were also included in the fetal brain scATAC-seq data). We identified 64 significant disease-cell type pairs (FDR < 5% for positive $\tau^*$ conditional on other annotations; Table 1, Table 2, Fig. 3B, Supplementary Data 16). When restricting to the 33 significant disease-cell type pairs corresponding to 8 cell types included in both scATAC-seq and scRNA-seq data, 20 of 33 were also significant in analyses of scATAC-seq data. The most significant enrichment was observed for excitatory neurons in intelligence, consistent with previous genetic

studies[21]. We also identified an enrichment of corticofugal projection neurons (CPN) in intelligence, which has not (to our knowledge) previously been reported in analyses of genetic data. CPN connect neocortex and the subcortical regions and transmits axons from the cortex. Imbalance in neuronal activity, particularly regarding excitability of CPNs, has been hypothesized to lead to deficits in learning and memory[54,55]. Recently[56] reported that NEUROD2 knockout mice display synaptic and physiological defects in CPN along with autism-like behavior abnormalities (where NEUROD2 is a transcription factor involved in early neuronal differentiation). CPN has previously been reported to be enriched in autism spectrum disorder (ASD) genes[57], we did not detect a significant ASD enrichment for CPN ($p$ = 0.056) or any other cell type (see Discussion).

**A**

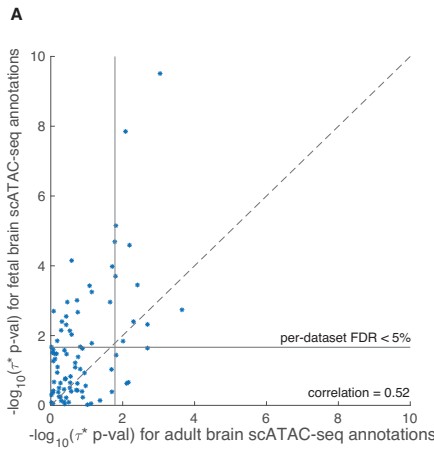
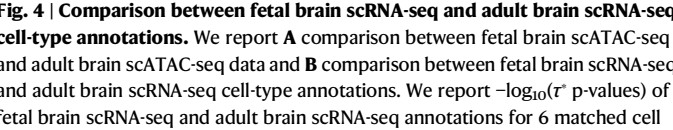

**B**

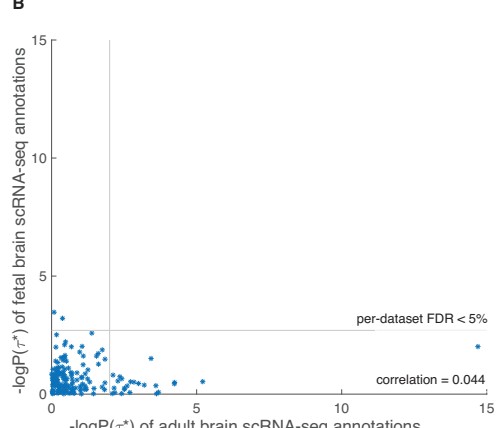

**Fig. 4 | Comparison between fetal brain scRNA-seq and adult brain scRNA-seq cell-type annotations.** We report **A** comparison between fetal brain scATAC-seq and adult brain scATAC-seq data and **B** comparison between fetal brain scRNA-seq and adult brain scRNA-seq cell-type annotations. We report $-\log_{10}(\tau^*$ p-values) of fetal brain scRNA-seq and adult brain scRNA-seq annotations for 6 matched cell types (astrocytes, endothelial cells, microglia, oligodendrocytes, excitatory neurons, inhibitory neurons), conditioning on the baseline model, union of open chromatin regions, and each other. Numeric results are found in Supplementary Data 18 and S19. Correlation among cell-type annotations is found in Supplementary Data 9.

We compared the results for 9 adult brain cell types included in both the scATAC-seq and scRNA-seq datasets (Fig. 3C and Supplementary Data 17). While scATAC-seq and scRNA-seq cell-type annotations for matched cell types were weakly correlated to each other ($r = 0.01–0.09$; Supplementary Data 9), the corresponding $-\log_{10}$(p-values) for positive $\tau^*$ were moderately correlated ($r = 0.25$), confirming the shared biological information. We observed more significant p-values for scATAC-seq than for scRNA-seq in these data sets, analogous our analyses of fetal brain data (see Discussion).

We compared the results for 3 cell types (astrocytes, inhibitory neurons, excitatory neurons) included in both fetal brain and adult brain scATAC-seq data sets (Fig. 4A and Supplementary Data 18). While fetal brain and adult brain cell-type annotations for matched cell types were weakly correlated to each other ($r = 0.00–0.01$), the corresponding $-\log_{10}$(p-values) for positive $\tau^*$ attained a moderately high correlation ($r = 0.52$), higher than the analogous correlations for scATAC-seq vs. scRNA-seq results ($r = 0.24$ for fetal brain, $r = 0.25$ for adult brain; see above). Interestingly, the enrichment in ADHD for fetal brain astrocytes (see above) was not observed for adult brain astrocytes ($p = 0.52$ for positive $\tau^*$, $p = 0.0065$ for difference in $\tau^*$ for adult brain astrocytes vs. fetal brain astrocytes). While astrocytes participate in defense against stress, energy storage, and tissue repair, they also mediate synaptic pruning (elimination of synaptosomes) during development[58]. Indeed, astrocytes in more mature stages of brain development were found to be less efficient at removing synaptosomes compared to younger, fetal astrocytes[59] (in both in vitro in pluripotent stem cells and in vivo mice), supporting a fetal brain-specific role of astrocytes in brain-related diseases and traits. We also determined that the enrichment in ADHD for fetal inhibitory neurons was not observed for adult brain inhibitory neurons ($p = 0.52$ for positive $\tau^*$, $p = 2.4 \times 10^{-4}$ for difference in $\tau^*$ for adult brain inhibitory neurons vs. fetal brain inhibitory neurons).

We observed little correlation between fetal brain and adult brain $-\log_{10}$(p-values) for positive $\tau^*$ in analyses of scRNA-seq data ($r = 0.044$; Fig. 4 and Supplementary Data 19), possibly due to the lower power of these analyses (particularly for fetal brain scRNA-seq) in the data sets that we analyzed (see Discussion).

We performed 5 secondary analyses. First, we analyzed enrichments of both scATAC-seq and scRNA-seq brain cell types in 6 control (non-brain-related) diseases and complex traits. As expected, we did not identify any significant enrichments (Supplementary Data 20 and Supplementary Data 21). Second, we repeated our disease heritability

enrichment analyses of scATAC-seq annotations while conditioning only on the baseline model (and not the union of open chromatin regions across all brain cell types). We identified 246 significant disease-cell type pairs, as compared to 168 significant disease-cell type pairs in our primary analysis (Figure S4A, Supplementary Data 22A). This underscores the importance of conditioning on the union of open chromatin regions across all cell types, a conservative step to ensure cell-type specificity. (However, in analyses of fetal brain scATAC-seq, we obtained similar results with or without additionally conditioning on the union of open chromatin regions across all brain cell types; Figure S4B, Supplementary Data 22B). Third, we performed gene set enrichment analysis using GREAT[47] for both scATAC-seq and scRNA-seq cell-type annotations from adult brain. As expected, we identified significant enrichments in relevant gene sets (Supplementary Data 23). Fourth, for the adult scRNA-seq data[14], we constructed annotations based on a ±100 kb window-based strategy (previously used in[8]) instead of brain-specific enhancer-gene links[7,30,31] (used in[22]). We identified only 28 significant trait-cell type pairs (Supplementary Data 24), vs. 64 using brain-specific enhancergene links. Fifth, we analyzed bulk chromatin data (7 chromatin marks) spanning 21 adult brain tissues[9] (age 27–85 years). We identified 1,710 significant disease-tissue-chromatin mark triplets spanning 26 of 28 brain-related diseases and traits (Supplementary Data 25). Once again, these results are included for completeness, but cannot achieve the same cell-type specificity as analyses of single-cell data.

## Discussion

We identified a rich set of disease-critical fetal and adult brain cell types by integrating GWAS summary association statistics from 28 brain-related diseases and traits with scATAC-seq and scRNA-seq data from 83 fetal and adult brain cell types[14–17]. We confirmed many previously reported disease-cell type associations, but also identified disease-cell type associations supported by known biology that were not previously reported in analyses of genetic data. We determined that cell-type annotations derived from scATAC-seq were particularly powerful in the data that we analyzed. We also determined that the disease-cell type associations that we identified can be either shared or specific across fetal vs. adult brain developmental stages.

We note 4 key distinctions between our work and previous studies identifying disease-critical tissues and cell types[4–8,10,12,16–19,21,22]. First, we explicitly compared results from scATAC-seq vs. scRNA-seq data in matched cell types. Although applications of single-cell data to identify

disease - critical cell types have largely prioritized analyses of scRNA-seq data[3], we determined that cell-type annotations derived from scATAC-seq were even more powerful in our analyses. This finding may be specific to limited power and reproducibility of scRNA-seq in the data that we analyzed, thus should not preclude further prioritization of scRNA-seq data. Second, we explicitly compared results for fetal and adult brain in matched cell types. We determined that concordance between fetal and adult brain scATAC-seq results ($r = 0.52$ for $-\log_{10}$(p-values) for positive $\tau^*$; Fig. 4A) was larger than concordance between fetal and adult brain scRNA-seq results ($r = 0.044$ for $-\log_{10}$(p-values) for positive $\tau^*$; Fig. 4); this cannot be explained by similarity between fetal and adult brain scATAC-seq cell-type annotations, which was low ($r = 0.00-0.01$). The simplest explanation for this result is the higher overall power of scATAC-seq annotations (e.g., 152 significant disease-fetal cell type pairs, reducing to 43 when restricting to cell types with both fetal and adult scATAC-seq data) vs. scRNA-seq annotations (e.g., 9 significant disease-fetal cell type pairs, reducing to 0 when restricting to cell types with both fetal and adult scRNA-seq data) in our analyses. However, disease-critical cell types were specific to fetal vs. adult brain developmental stages in some scATAC-seq analyses, such as the enrichment of fetal astrocytes in ADHD. Third, we rigorously conditioned on a broad set of other functional annotations, a conservative step to ensure cell-type specificity that was included in recent unpublished work[33,34], but not included in[17,19]. In particular, for scATAC-seq annotations, we conditioned on the union of open chromatin regions across all brain cell types in each data set analyzed, in addition to the baseline model[11]. For scRNA-seq annotations, we conditioned on the union of brain-specific enhancer-gene links across all genes analyzed, in addition to the baseline model[11]. Fourth, in analyses of scRNA-seq data, we constructed annotations using brain-specific enhancer-gene links[7,30,31] (used in[22]), an emerging approach that is more powerful than conventional window-based strategies for linking SNPs to genes.

Our findings have implications for improving our understanding of how cell-type specificity impacts disease risk. Better understanding disease-critical cell types is crucial to characterizing disease mechanisms underlying cell type specificity and developing new therapeutics[3]. To this end, the disease-cell type associations that we identified can help guide functional follow-up experiments (e.g., Perturb-seq[60], saturation mutagenesis[61], and CRISPR-Cas9 cytosine base editor screen[62]) to study cellular mechanisms of specific loci or genes underlying disease. In addition, our results highlight the benefits of analyzing data from different sequencing platforms and different developmental stages to identify disease-critical cell types. This motivates the prioritization of technologies that simultaneously profile ATAC and RNA expression such as SHARE-seq[63], as well as continuing efforts to profile the developing human brain[34].

We note several limitations of our work. First, although annotations derived from scATAC-seq generally outperformed annotations derived from scRNA-seq in the data that we analyzed, we caution that we are unable to draw any universal conclusions about which technology is most useful, as our findings may be impacted by the particularities of the data sets that we analyzed. However, we note that for both fetal and adult brain, the scRNA-seq data that we analyzed had larger numbers of donors and nuclei sequenced vs. the scATAC-seq data. Second, our resolution in identifying disease-critical cell types is fundamentally limited by the resolution of annotated cell types in the single-cell data that we analyzed; in particular, rare but biologically important cell types may be poorly represented in these data sets. Emerging approaches that assess disease enrichment at the level of individual cells rather than annotated cell types[64,65] could overcome this limitation. Third, despite our rigorous efforts to condition on a broad set of functional annotations, we are unable to conclude that the disease-critical cell types that we identify are biologically causal; it may

often be the case that they tag a biologically causal cell type that is not included in the data that we analyzed. This motivates further research on methods for discriminating closely related cell types[18] and fine-mapping causal cell types (analogous to research on fine-mapping disease variants[66] and disease genes[67]). Fourth, we failed to identify any significant cell types for 4 diseases/traits (autism, anorexia, ischemic stroke, and Alzheimer's disease), possibly due to limited GWAS power and/or disease heterogeneity. Fifth, we did not identify a few well-known disease-cell type associations (e.g., microglia for Alzheimer's disease), potentially due to our conservative assessment of enrichments and stringent multiple testing corrections. Despite these limitations, the disease-cell type associations that we identified have high potential to improve our understanding of the biological mechanisms of complex disease.

## Methods
### 28 distinct brain-related diseases and traits
We considered 146 sets of GWAS summary association statistics, including 83 traits from the UK Biobank and 63 traits from publicly available sources, with z-scores for total SNP-heritability of at least 6 (computed using S-LDSC with the baseline-LD (v.2.2) model); while we use the baseline-LD model for this specific purpose of computing z-scores, as noted below, we used the baseline model in estimating the heritability enrichment. We selected 31 brain-related traits based on previous studies[8,17,21,22,68]. We removed 3 traits (with lower SNP-heritability z-score) that had a genetic correlation of at least 0.9 with at least one of these 31 traits, retaining a final set of 28 distinct brain-related traits (including 7 traits from the UK Biobank) (Supplementary Data 1). The genetic correlations among the 28 traits are reported in Supplementary Data 2. Genetic correlations ($r$) are estimated from GWAS summary statistics using cross-trait S-LDSC[69].

We additionally analyzed 6 distinct control (non-brain-related) traits: coronary artery disease, bone mineral density, rheumatoid arthritis, type 2 diabetes, sunburn occasion, and breast cancer. These 6 traits had similar sample sizes and SNP-heritability z-scores as the 28 brain-related traits.

### Ethical approval
The ethical approval and ethical compliance of the 4 published data sets is as follows:

For the Domcke et al.[16] and Cao et al.[15] data set, human fetal tissues (89 to 125 days estimated post-conceptual age) were obtained by the University of Washington Birth Defects Research Laboratory (BDRL) under a protocol approved by the University of Washington Institutional Review Board.

For the Corces et al.[17] data set, primary brain samples were acquired postmortem with institutional review board-approved informed consent from Stanford University, the University of Washington or Banner Health. For the Velmeshev et al.[14] data set, de-identified snap-frozen post-mortem tissue samples from ASD and epilepsy patients and control donors without neurological disorders were obtained and approved by University of Maryland Brain Bank Institutional Review Board through the NIH NeuroBioBank.

### Genomic annotations and the baseline model
We define a binary genomic annotation as a subset of SNPs in a pre-defined reference panel. We restrict our analysis to SNPs with a minor allele frequency (MAF) $\geq 0.5\%$ in 1000 Genomes[28] (see Web resources).

The baseline model[32] (v.1.2; see Supplementary Data 3) contains 53 binary functional annotations (see Web resources). These annotations include genomic elements (e.g., coding, enhancer, UTR), regulatory elements (e.g., histone marks), and evolutionary constraint. We included the baseline model, consistent with[8,36], when assessing the heritability enrichment of the cell-type annotations.

## Single-cell ATAC-seq data

We considered single-cell ATAC-seq data for fetal brains from Domcke et al.[16] (donor size = 26; 15 males and 11 females) and adult brains (isocortex, striatum, hippocampus, and substantia nigra) of cognitively healthy individuals from Corces et al.[17] (donor size = 10; 4 males and 6 females). (Based on these sex distributions, we believe it is unlikely that the sex distribution of donors substantially impacted our findings.) We used the chromatin accessible peaks for each cell type without modifications (see Web resources). In short, these peaks refer to MACS2[28] peak regions, excluding the ENCODE blacklist regions. For the Domcke et al. data, authors called peaks on each tissue sample and then generated a masterlist of all peaks across all samples and generated the cell-type-specific peaks using Jensen-Shannon divergence[70]. To further ensure the cell-type specificity, we used the union of per-dataset open chromatin regions across all cell types as the background annotation in the S-LDSC conditional analysis.

## Single-cell RNA-seq data analyzed

We considered single-cell RNA-seq data for fetal brains from Cao et al.[15] (donor size = 28; 14 males and 14 females) and single-cell RNA-seq data for non-fetal brains (prefrontal cortex and anterior cingulate cortex) from Velmeshev et al.[14] (donor size = 31; 24 males and 7 females). (Based on these sex distributions, and the fact that the Velmeshev et al. data produced an intermediate number of significant disease-cell type pairs (64/476; Table 1), we believe it is unlikely that the sex distribution of donors substantially impacted our findings. For Cao et al. data, we processed data from three brain-related organs: cerebellum, cerebrum, and eye. For each data set, we used the sc-linker pipeline[22] to construct probability scores annotating SNPs linked to specifically expressed genes in a given cell type[8] (compared to other brain cell types) using brain-specific enhancer-gene links[7,22,30,31]. Complete details are provided in ref. 22. In brief, we downloaded metadata for each cell including the total number of reads and sample ID. We then transformed each expression matrix to log2(TP10K + 1) units. We performed a dimensionality reduction using a principal component analysis with the top 2000 highly variable genes, batch correction using Harmony[71], and applied the Leiden graph clustering method[72]. To obtain specifically expressed gene scores for each cell type, we applied a non-parametric Wilcoxon rank-sum test between gene expression from focal cell type vs. gene expression in other cell types; specific expression was assessed relative to all brain cell types. We transformed the per-gene p-value for specific expression to a probabilistic specifically expressed gene score between 0 and 1, by applying min-max normalization on $-2\log$(p-value), indicating a relative importance of each gene in each cellular process. To construct probability scores annotating SNPs linked to specifically expressed genes from specifically expressed gene scores, we employed an enhancer-gene linking strategy from the union of the Roadmap[7] and Activity-By-Contact (ABC[30,31]) strategies. Because we focused on brain-related traits, we used brain-specific enhancer-gene links. Probability scores annotating SNPs linked to specifically expressed genes were defined based on the maximum specifically expressed gene score among genes linked to a SNP (or 0 when no genes are linked to a SNP).

## Enrichment and $\tau^*$ metrics

We used stratified LD score regression (S-LDSC[11,32]) to assess the contribution of an annotation to disease and complex trait heritability.

Let $a_{cj}$ represent the (binary or probabilistic) annotation value of the SNP $j$ for the annotation $c$. S-LDSC assumes the variance of per normalized genotype effect sizes is a linear additive contribution to the annotation $c$:

$$Var\left(\beta_j\right) = \sum_c a_{cj}\tau_c \tag{1}$$

where $Var(\beta_j)$ is the variance of effect sizes $\beta_j$ of standardized genotype for each $SNP_j$, $\tau_c$ is the per-SNP contribution of the annotation $c$. We note that each scATAC-seq analysis includes 55 annotations (1 focal cell-type-specific annotation + 53 baseline model annotations + 1 annotation consisting of the union of open chromatin regions across all brain cell types in the scATAC-seq data set being analyzed) and each scRNA-seq analysis includes 55 annotations (1 focal cell-type-specific annotation + 53 baseline model annotations + 1 annotation consisting of the union of brain-specific enhancer-gene links across all genes analyzed).

S-LDSC estimates $\tau_c$ using the following equation:

$$E\left[\chi_j^2\right] = N\sum_c l(j,c)\tau_c + 1 \tag{2}$$

where $\chi_j^2$ is the chi-square association statistic for SNP $j$, $N$ is the sample size of the GWAS and $l(j,c)$ is the LD score of the SNP $j$ to the annotation $c$. The LD score is computed as follows: $l(j,c) = \sum_k a_{ck}r_{jk}^2$ where $r_{jk}$ is the correlation between the SNPs $j$ and $k$.

We used two metrics to assess the informativeness of an annotation. First, the standardized effect size ($\tau^*$), the proportionate change in per-SNP heritability associated with a one standard deviation increase in the value of the annotation (conditional on all the other annotations in the model), is defined as follows:

$$\tau_c^* = \frac{\tau_c sd(a_c)}{h_g^2/M} \tag{3}$$

where $sd(a_c)$ is the standard deviation of the annotation c, $h_g^2$ is the estimated SNP-heritability, and $M$ is the number of variants used to compute $h_g^2$ (in our experiment, $M$ is equal to 5,961,159, the number of common SNPs in the reference panel). The significance for the effect size for each annotation, as mentioned in previous studies[32,68,73], is computed as ($\frac{\tau^*}{se(\tau^*)} \sim N(0,1)$), assuming that $\frac{\tau^*}{se(\tau^*)}$ follows a normal distribution with zero mean and unit variance.

Second, enrichment of the binary and probabilistic annotation is the fraction of heritability explained by SNPs in the annotation divided by the proportion of SNPs in the annotation, as shown below:

$$Enrichment = \frac{\%h_g^2(C)}{\%SNP(C)} = \frac{\frac{h_g^2(C)}{h_g^2}}{\frac{\sum_j a_{jc}}{M}} \tag{4}$$

where $h_g^2(C)$ is the heritability captured by the $c$-th annotation. When the annotation is enriched for trait heritability, the enrichment is > 1; the overlap is greater than one would expect given the trait heritability and the size of the annotation. The significance for enrichment is computed using the block jackknife as mentioned in previous studies[8,11,68,73]). The key difference between enrichment and $\tau^*$ is that $\tau^*$ quantifies effects that are unique to the focal annotation after conditioning on all the other annotations in the model, while enrichment quantifies effects that are unique and/or non-unique to the focal annotation.

We used European samples in 1000G[28] as reference SNPs and HapMap 3[74] SNPs as regression SNPs (see Web resources). We excluded SNPs with marginal association statistics > 80 and SNPs in the major histocompatibility complex region. In all our analyses, we used the p-value of $\tau^*$ as our primary metric to estimate the effect sizes conditional on known annotations (by including the baseline model as recommended previously[8,36]). We excluded trait-annotation pairs with negative $\tau^*$, consistent with previous studies[16,32,60]. We assessed the statistical significance of trait-cell type associations based on per-dataset FDR < 5% (more conservative than[16]), aggregating across 28 brain-related traits and all cell types in the dataset (or aggregating across 6 control traits and all cell types in the dataset, in analyses of control traits). As we expect no enrichments of brain cell types in these 6 control traits, we controlled FDR separately from the analysis of brain traits.

## Gene set enrichment analysis using GREAT

We performed gene set enrichments on each cell-type annotations for the gene ontology (GO) biological process, cellular component, and molecular function. We used GREAT[47] (v.4.0.4) with its default setting, where each gene is assigned a regulatory domain (for proximal: 5 kb upstream, 1 kb downstream of the TSS; for distal: up to 1 Mb). Because annotations from the scRNA-seq were probabilistic, we limited to regions with gene membership probability >= 0.8 for gene set enrichment analysis. We used all regions for the scATAC-seq annotations as an input. We defined significant results as those with the FDR-corrected one-tailed binomial test $p$-value < 0.05.

## Reporting summary

Further information on research design is available in the Nature Portfolio Reporting Summary linked to this article.

## Data availability

Cell-type annotations generated for primary analyses of disease-critical cell types in this study: https://alkesgroup.broadinstitute.org/LDSCORE/Kim_ATAC/. GWAS summary statistics used to assess disease/trait heritability enrichment: https://alkesgroup.broadinstitute.org/sumstats_formatted/. Domcke et al.[16] data used to identify disease-critical fetal brain cell types using scATAC-seq: https://atlas.brotmanbaty.org/bbi/human-chromatin-during-development/. Cao et al.[15] data used to identify disease-critical fetal brain cell types using scRNA-seq: https://atlas.brotmanbaty.org/bbi/human-gene-expression-during-development/. Corces et al.[17] data used to identify disease-critical adult brain cell types using scATAC-seq: http://epigenomegateway.wustl.edu/legacy/?genome=hg38. &session=drS3o1n4kJ. Velmeshev et al.[14] data used to identify disease-critical adult brain cell types using scRNA-seq: https://autism.cells.ucsc.edu/. Baseline (v.1.2) annotations used as additional annotations when running S-LDSC: https://data.broadinstitute.org/alkesgroup/LDSCORE/. 1000 Genomes Project Phase 3 data used as reference data when running S-LDSC: ftp://ftp.1000genomes.ebi.ac.uk/vol1/ftp/release/20130502 Source data are provided with this paper.

## Code availability

The source code used to generate cell-type annotations for primary analyses of disease-critical cell types in this study are available at https://github.com/buutrg/Kim_ATAC_code. S-LDSC software used to assess disease/trait heritability enrichment: https://github.com/bulik/ldsc. GREAT (Genomic Regions Enrichment of Annotations Tool) software used to perform gene set enrichment analysis: http://great.stanford.edu/.

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

## Acknowledgements

We are grateful to Tiffany Amariuta, Katie Siewert, Martin Zhang, and Huwenbo Shi for their helpful discussions. This research was funded by NIH grants U01 HG009379, U01 MH119509, R01 MH101244, R37 MH107649, R01 MH115676, R01 MH109978, U01 HG012009, and R01 HG006399. S.S.K. was supported by the NIH NHGRI award F31HG010818. This research was conducted using the UK Biobank Resource under Application 16549. K.K.Dey is funded by R00HG012203, P30 CA008748, and the Josie Robertson Investigators Program.

## Author contributions

S.S.K. and A.L.P. designed experiments. S.S.K. performed experiments. K.J. and K.K.D. processed scRNA-seq data. A.Z.S assisted in processing scATAC-seq data. B.T., S.R., M.K., and A.L.P. provided guidance and feedback on analyses. S.S.K., B.T., and A.L.P. wrote the manuscript with the assistance from all authors.

## Competing interests

The authors declare no competing interests.
