## [Peer Review File · Nature Communications]

Leveraging single-cell ATAC-seq and RNA-seq to identify disease-critical fetal and adult brain cell typesREVIEWER COMMENTS

Reviewer #1 (Remarks to the Author):

The authors present results from analyses of scRNAseq and scATACseq data obtained from fetal (brain samples at the developmental stage) and adult samples (brain samples after the developmental stage) with the goal to identify cell types associated with brain diseases/traits. The analyses are based on well-established methods, and the results support the conclusions. Specifically, they identified several cell type-disease associations by analyzing the single-cell data aggregated from four different resources (this data process appears to be quite a complicated task). The new findings include associations between photoreceptor cells with insomnia and major depressive disorder.

One interesting observation made by the proposed analysis was that compared to scRNAseq data, scATACseq data was more helpful in identifying associations with cell types and diseases. This observation seems counterintuitive since the scRNAseq data are considered the final product of the transcription process. Therefore, one may think that scRNAseq data should provide more biological insights. This observation may be due to the difficulties and complexities of the annotation process or can be influenced by the quality of samples or the sample acquisition process. On the other hand, the process of generating dichotomous annotations from scATACseq data is more intuitive and easier to understand.

Comments:

The Methods section seems to be missing some critical parts needed to understand how they obtain probability scores based on scRNAseq data. The analysis results showed a strong correlation between the association analysis results using scATACseq annotations and scRNAseq annotations. Though based on this observation, I am curious what would happen if the authors generated a binary annotation for scRNAseq data by converting all positive probability scores to 1.

The definition of heritability enrichment described in the results section requires revision in the current version of the manuscript. In the methods section, the denominator of the heritability enrichment score appears to be the sum of the annotation values, not the number of annotations greater than zero.

What does “r” in the results part represent? It appears to be a genetic correlation estimated from the GWAS summary statistics.

I think the authors can add the results of testing the normality assumption of $\tau^*/se(\tau^*)$, which will increase the reliability of the proposed analysis in the paper.

Is it true that $SD(C)=SD(a_c)$?

Reviewer #2 (Remarks to the Author):

Kim and colleagues present an analysis identifying disease-critical cell types by integrating genome wide association study (GWAS) statistics and single cell genomic data. The authors focused on 28 brain-related diseases and traits and identified their relevant cell types from two modalities of single cell profiling: including single-cell RNA-seq (scRNA-seq) and single-cell ATAC-seq (scATAC-seq). These single cell data involved not only fetal but also adult brain cells, so that the disease-critical cell types can be identified and compared across different facets of data types and developmental stages. The enrichment of cell type is generally more significant and has a higher specificity for the analysis using scATAC-seq compared to scRNA-seq, although the enrichments were moderately consistent.

Overall, this study provides a comprehensive discovery of cell types that may be relevant to 28 brain-related diseases and traits by using both scATAC-seq and scRNA-seq. The disease-cell type associations include those that are supported by known biology, but also others that were not previously reported in analyses of genetic data, which may enable new insights through functional follow up studies. The most exciting part of this work is the comparison of analyses between different modalities for the same disease in matched cell types. This analysis highlights advantages and limitations of using different genomic data to functionally discern disease-relevant variants. I have a few comments for improvement:

Major Comments:

1. The analyses of scATAC-seq between fetal and adult brain yield more concordant enrichments across cell types than that of scRNA-seq, it will be great to discuss potential reasons why this may be the case.
2. All the trait-cell type pairs are identified based on pseudo-bulk data for both scRNA-seq and scATAC-seq. A recent method (DOI: 10.1038/s41587-022-01341-y) enables enrichment of cell types at single-cell resolution from scATAC-seq. In light of these recent advances, the authors should discuss how using annotated cell types in single cell data could limit the resolution of their findings and discuss how emerging approaches, such as the paper highlighted above, could overcome these limitations.
3. The authors state that “mental illness is not only a consequence but also a cause of visual impairment.” There is no supportive data for making this statement in the paper and it should be removed.

Minor Comments:

1. The abbreviation “resp.” in Abstract should be defined.
2. The citation format should be fixed, such as the sentence (...consistent with recent unpublished work^{32, 33}, but different from ref. 19, 17...) in the first section of Results and the sentence (...unpublished work^{32,33}, but not included in ref. 19, 17.) in Discussion.
3. The title only highlights the use of scATAC-seq in disease-critical cell type identification, but the paper includes analysis of scRNA-seq too.

4. This sentence in the Discussion is not clear: “This finding may be specific to the data that we analyzed, and should not preclude further prioritization of scRNA-seq data, but does motivate further prioritization of scATAC-seq data.” It could be modified to make a clearer conclusion.

Response to reviewers for NCOMMS-22-17004 (Kim et al.)

Reviewer #1:

The authors present results from analyses of scRNAseq and scATACseq data obtained from fetal (brain samples at the developmental stage) and adult samples (brain samples after the developmental stage) with the goal to identify cell types associated with brain diseases/traits. The analyses are based on well-established methods, and the results support the conclusions. Specifically, they identified several cell type-disease associations by analyzing the single-cell data aggregated from four different resources (this data process appears to be quite a complicated task). The new findings include associations between photoreceptor cells with insomnia and major depressive disorder.

One interesting observation made by the proposed analysis was that compared to scRNAseq data, scATACseq data was more helpful in identifying associations with cell types and diseases. This observation seems counterintuitive since the scRNAseq data are considered the final product of the transcription process. Therefore, one may think that scRNAseq data should provide more biological insights. This observation may be due to the difficulties and complexities of the annotation process or can be influenced by the quality of samples or the sample acquisition process. On the other hand, the process of generating dichotomous annotations from scATACseq data is more intuitive and easier to understand.

We thank the reviewer for suggesting that our work is informative and for stating that our conclusions are supported by the results.

Comments:

1. The Methods section seems to be missing some critical parts needed to understand how they obtain probability scores based on scRNAseq data. The analysis results showed a strong correlation between the association analysis results using scATACseq annotations and scRNAseq annotations. Though based on this observation, I am curious what would happen if the authors generated a binary annotation for scRNAseq data by converting all positive probability scores to 1.

The reviewer has raised two requests: (i) The Methods section requires more explanation of how probability scores were obtained from scRNA-seq data; and (ii) What happens if we analyze binary annotations for scRNA-seq data by converting all positive probability scores to 1. We address each of these requests in turn:

(i) The Methods section requires more explanation of how probability scores were obtained from scRNA-seq data.

We recognize that it is our responsibility to provide a clear exposition in describing how probability scores were obtained from scRNA-seq data.

First, we have updated both the *Overview of methods* subsection of the Results section (p.4) and the *Single-cell RNA-seq data analyzed* subsection of the Methods section (p.13) to clarify that probability scores were obtained using the sc-linker pipeline of Jagadeesh*,Dey* et al. 2022 Nat Genet (ref. 22; formerly Jagadeesh*,Dey* et al. biorxiv).

Second, we have modified the text in the *Single-cell RNA-seq data analyzed* subsection of the Methods section (p.13-14) describing the sc-linker pipeline, in an effort to improve clarity. We believe that the modified text (roughly 1/2 page of text) is a sufficient amount of text to describe a pipeline that is now published (Jagadeesh*,Dey* et al. 2022 Nat Genet; ref. 22), but we are open to further expanding any part of this description if that is recommended.

(ii) What happens if we analyze binary annotations for scRNA-seq data by converting all positive probability scores to 1.

As requested by the reviewer, we performed a new analysis in which we constructed binary annotations for scRNA-seq data by converting all positive probability scores to 1. We determined that this produced results that were similar to but slightly worse than our primary analysis involving probability scores (τ^* regression slope = 0.677). We report these new results in a new Figure S3. We have updated the *Identifying disease-critical cell types using fetal brain data* subsection of the Results section (p.6) to discuss these results, citing Figure S3.

In conjunction with this analysis, we believe it is of interest to report the distribution of nonzero probability scores. We determined that most nonzero probability scores are either close to 0 or close to 1. We report these distributions in a new Figure S4. The fact that binarizing the probability scores produces slightly worse results implies that nonzero probability scores that are close to 0 are less informative than nonzero probability scores that are close to 1. We have updated the *Identifying disease-critical cell types using fetal brain data* subsection of the Results section (p.6-7) to discuss these results, citing Figure S4.

2. The definition of heritability enrichment described in the results section requires revision in the current version of the manuscript. In the methods section, the denominator of the heritability enrichment score appears to be the sum of the annotation values, not the number of annotations greater than zero.

We thank the reviewer for pointing this out, and regret the error. We have changed this text in the *Overview of methods* subsection of the Results section (p.4) from “the proportion of heritability explained divided by the proportion of annotated SNPs” to “the proportion of heritability explained divided by the annotation size, which is defined as the average annotation value for probabilistic annotations.”

3. What does “r” in the results part represent? It appears to be a genetic correlation estimated from the GWAS summary statistics.

The reviewer is correct that “r” denotes a genetic correlation estimated from the GWAS summary statistics. We have updated the *28 distinct brain-related diseases and traits* subsection of the Methods section (p.12) to clarify this point.

4. I think the authors can add the results of testing the normality assumption of $\frac{\tau^*}{se(\tau^*)}$, which will increase the reliability of the proposed analysis in the paper.

We agree that it is of interest to test the normality of assumption of $\frac{\tau^*}{se(\tau^*)}$. We have now done this, by investigating the distribution of p-values for nonzero τ^* (based on the normality assumption) in analyses of both scATAC-seq and scRNA-seq fetal brain cell types across 6 non-brain-related diseases and complex traits, for which results are expected to be null. Q-Q plots confirm a null distribution of P-values (new Figure S2). We have updated the *Identifying disease-critical cell types using fetal brain data* subsection of the Results section (p.6) to discuss these results, citing Figure S2.

5. Is it true that $sd(C) = sd(a_c)$?

The reviewer is correct that $sd(C)$ in Equation 3 refers to the standard deviation of the annotation (i.e. $sd(a_c)$). Accordingly, we have changed $sd(C)$ to $sd(a_c)$ in Equation 3 (*Enrichment and τ^* metrics* subsection of the Methods section, p.14).

Reviewer #2:

Kim and colleagues present an analysis identifying disease-critical cell types by integrating genome wide association study (GWAS) statistics and single cell genomic data. The authors focused on 28 brain-related diseases and traits and identified their relevant cell types from two modalities of single cell profiling: including single-cell RNA-seq (scRNA-seq) and single-cell ATAC-seq (scATAC-seq). These single cell data involved not only fetal but also adult brain cells, so that the disease-critical cell types can be identified and compared across different facets of data types and developmental stages. The enrichment of cell type is generally more significant and has a higher specificity for the analysis using scATAC-seq compared to scRNA-seq, although the enrichments were moderately consistent.

Overall, this study provides a comprehensive discovery of cell types that may be relevant to 28 brain-related diseases and traits by using both scATAC-seq and scRNA-seq. The disease-cell type associations include those that are supported by known biology, but also others that were not previously reported in analyses of genetic data, which may enable new insights through functional follow up studies. The most exciting part of this work is the comparison of analyses between different modalities for the same disease in matched cell types. This analysis highlights advantages and limitations of using different genomic data to functionally discern disease-relevant variants.

We thank the reviewer for providing an accurate summary of our work and highlighting exciting aspects of our work.

Major comments:

1. The analyses of scATAC-seq between fetal and adult brain yield more concordant enrichments across cell types than that of scRNA-seq, it will be great to discuss potential reasons why this may be the case.

The reviewer is correct that concordance between fetal and adult brain scATAC-seq results ($r=0.52$ for $-\log_{10}(\text{p-values})$ for positive τ^{\square} ; Fig. 2D) was larger than concordance between fetal and adult brain scRNA-seq results ($r=0.044$ for $-\log_{10}(\text{p-values})$ for positive τ^{\square} ; Fig. S5 (formerly Fig. S2)). We have updated the Discussion section (p. 10) to discuss this point.

2. All the trait-cell type pairs are identified based on pseudo-bulk data for both scRNA-seq and scATAC-seq. A recent method (DOI: 10.1038/s41587-022-01341-y) enables enrichment of cell types at single-cell resolution from scATAC-seq. In light of these recent advances, the authors should discuss how using annotated cell types in single cell data could limit the resolution of their findings and discuss how emerging approaches, such as the paper highlighted above, could overcome these limitations.

We agree that emerging approaches that assess disease enrichment at the level of individual cells rather than annotated cell types could overcome this limitation. We have updated the Discussion section (p.11, citing the paper of Yu et al. 2022 Nat Biotechnol mentioned by the reviewer) to discuss this point.

3. The authors state that “mental illness is not only a consequence but also a cause of visual impairment.” There is no supportive data for making this statement in the paper and it should be removed.

We have removed this sentence from the *Identifying disease-critical cell types using fetal brain data* subsection of the Results section (p.5).

Minor comments:

1. The abbreviation “resp.” in Abstract should be defined.

We have changed “resp.” to “respectively” in the Abstract (p.2).

2. The citation format should be fixed, such as the sentence (...consistent with recent unpublished work^{32, 33}, but different from ref. 19, 17...) in the first section of Results and the sentence (...unpublished work^{32,33}, but not included in ref. 19, 17.) in Discussion.

As requested by the reviewer, we have deleted all instances of “ref.” from the text, e.g. “consistent with recent unpublished work^{33,34}, but different from ^{17,19}” in the *Overview of methods* subsection of the Results section (p.4).

3. The title only highlights the use of scATAC-seq in disease-critical cell type identification, but the paper includes analysis of scRNA-seq too.

We have updated the title of the paper to “Application of single-cell ATAC-seq and RNA-seq to identify disease-critical fetal and adult brain cell types” (p.1).

4. This sentence in the Discussion is not clear: “This finding may be specific to the data that we analyzed, and should not preclude further prioritization of scRNA-seq data, but does motivate further prioritization of scATAC-seq data.” It could be modified to make a clearer conclusion.

We have updated this sentence in the Discussion section (p.10) to “This finding may be specific to the data that we analyzed, thus should not preclude further prioritization of scRNA-seq data.”

REVIEWERS' COMMENTS

Reviewer #2 (Remarks to the Author):

The authors have adequately addressed all of my concerns.

Reviewer #3 (Remarks to the Author):

The authors have integrated GWAS statistics with snRNA and snATAC-seq techniques to uncover new associations between diseases and cell types specific to the brain, both in fetal and adult contexts. Their analyses were conducted using publicly available and preprocessed data, employing established methods.

This study utilized high-quality ATAC-seq data from different brain regions, which had been previously published. While regional heterogeneity may impact the results concerning disease-cell type associations, a reasonable annotation correlation was observed between the two compared scATAC-seq datasets (Figure 2D).

Interestingly, the authors discovered that scATAC-seq data provides more informative and consistent results than scRNA-seq for identifying disease-critical cell types within these datasets. However, this conclusion is challenged by the limited power and reproducibility of the scRNA-seq modality, as pointed out by Reviewer 2.

To enhance reproducibility, it would be beneficial to include the equations of the model used in their analysis (conditioned on the broad set of open chromatin regions), and provide definitions for all variables used in the LD score equation. Information regarding the subjects' sex and its potential impact on the results are currently missing. Lastly, the code should be made accessible through a public repository.

Reviewer #4 (Remarks to the Author):

Kim and colleagues devised a robust statistical analysis method that leverages knowledge from GWAS to deduce disease associations specific to cell types in scRNA-seq or scATAC-seq datasets. They applied their approach to published scRNA-seq and scATAC-seq datasets, which span various developmental stages, brain regions, and sequencing batches. By integrating GWAS and single-cell profiling analyses, they identified cell types in the fetal or adult human brains that exhibit strong associations with neurological or psychiatric disorders. Moreover, they discovered that using their methods, scATAC-seq datasets are qualitatively more sensitive in identifying cell type-specific disease associations compared to scRNA-seq datasets, despite a general consistency in results between the two. Overall, their approach provides a way to bridge the gap between a multitude of GWAS targets and single-cell profiling datasets, which could enhance our understanding of disease mechanisms at the cellular level.

In general, I agree with the comments from Reviewer #1 regarding the original manuscript. Reviewer #1 requested a more comprehensive explanation in the Methods section about how the authors derived their probability scores from scRNA-seq datasets, and also asked for the results of employing an alternative analytic strategy (Question #1). The reviewer also identified several errors in the manuscript related to the calculation of enrichment (Questions #2 and #5) and correlation (Question #3). Additionally, they suggested a method to enhance the statistical reliability of the analyses presented in the original manuscript (Question #4).

The authors' response has adequately addressed all the queries and concerns raised by Reviewer #1.

Response to reviewers for NCOMMS-22-17004A (Kim et al.)

Reviewer #2:

The authors have adequately addressed all of my concerns.

We thank the reviewer for stating that all of their concerns were addressed in the revised manuscript.

Reviewer #3:

The authors have integrated GWAS statistics with snRNA and snATAC-seq techniques to uncover new associations between diseases and cell types specific to the brain, both in fetal and adult contexts. Their analyses were conducted using publicly available and preprocessed data, employing established methods.

We thank the reviewer for providing an accurate summary of the manuscript.

This study utilized high-quality ATAC-seq data from different brain regions, which had been previously published. While regional heterogeneity may impact the results concerning disease-cell type associations, a reasonable annotation correlation was observed between the two compared scATAC-seq datasets (Figure 2D).

We thank the reviewer for affirming the moderately high correlation ($r = 0.52$) between results for the fetal brain and adult brain scATAC-seq data sets (Figure 2D).

Interestingly, the authors discovered that scATAC-seq data provides more informative and consistent results than scRNA-seq for identifying disease-critical cell types within these datasets. However, this conclusion is challenged by the limited power and reproducibility of the scRNA-seq modality, as pointed out by Reviewer 2.

We agree with the reviewer that the higher informativeness of scATAC-seq vs. scRNA-seq for identifying disease-critical cell types within these data sets may be due to the limited power and reproducibility of scRNA-seq in the specific data sets that we analyzed. We previously modified the wording in the Discussion section pertaining to this point (p.10) based on a comment from Reviewer #2 (who has now stated that all of their concerns have been addressed). We have now further modified the wording in the Discussion section pertaining to this point (p.10) to specifically mention the limited power and reproducibility of scRNA-seq in the specific data sets that we analyzed, per the reviewer's comment.

To enhance reproducibility, it would be beneficial to include the equations of the model used in their analysis (conditioned on the broad set of open chromatin regions) ...

The equations of the S-LDSC model are provided as Equation 1 and Equation 2 in the Methods section (p.14). Our interpretation of the reviewer comment is that Equation 1 and Equation 2 are not sufficiently clear because those equations do not make clear which

annotations are included in the model in a particular analysis. (For example, as noted by the reviewer, the scATAC-seq analysis is conditioned on the broad set of open chromatin regions.) That information is summarized in the *Overview of methods* subsection of the Results section (p.4), but we agree that it is appropriate to provide detailed information about this in the text accompanying Equation 1 and Equation 2. Thus, we have now updated the text accompanying Equation 1 and Equation 2 in the Methods section (p.14) to clarify that:

-- Each scATAC-seq analysis includes 55 annotations: 1 focal cell-type-specific annotation + 53 baseline model annotations + 1 annotation consisting of the union of open chromatin regions across all brain cell types in the scATAC-seq data set being analyzed.

-- Each scRNA-seq analysis includes 55 annotations: 1 focal cell-type-specific annotation + 53 baseline model annotations + 1 annotation consisting of the union of brain-specific enhancer-gene links across all genes analyzed.

... and provide definitions for all variables used in the LD score equation.

The equations of the S-LDSC model are provided as Equation 1 and Equation 2 in the Methods section (p.14). We have now:

-- updated the text accompanying Equation 1 to provide the definition of $Var(\beta_j)$, which was previously missing.

-- updated the text accompanying Equation 2 to provide a definition of χ^2_j and to fix an error in the definition of $l(j,c)$, changing $P_k a_{ck} r_{jk}^2$ to $\sum_k a_{ck} r_{jk}^2$. We regret the error.

Information regarding the subjects' sex and its potential impact on the results are currently missing.

We have added this information to the "Single-cell ATAC-seq data" and "Single-cell RNA-seq data" subsections of the Methods section (p.13). Briefly, the Domcke et al. fetal brain scATAC-seq data (donor size = 26) included 15 males and 11 females; the Corces et al. adult brain scATAC-seq data (donor size = 10) included 4 males and 6 females; the Cao et al. fetal brain scRNA-seq data (donor size = 28) included 14 males and 14 females; the Velmeshev et al. non-fetal brain scRNA-seq data (donor size = 31) included 24 males and 7 females; and we have noted that based on this information, and the fact that the Velmeshev et al. non-fetal brain scRNA-seq data produced an intermediate number of significant disease-cell type pairs (64/476; Table 1), we believe it is unlikely that the sex distribution of donors substantially impacted our findings.

Lastly, the code should be made accessible through a public repository.

We have made the source code publicly available at https://github.com/buutrg/Kim_ATAC_code as indicated in the Code Availability section (p.16).

We also have made the cell-type annotations publicly available under a requester-pays model at https://alkesgroup.broadinstitute.org/LDSCORE/Kim_ATAC/, as indicated in the Data Availability section (p.16). (Previously, both source code and cell-type annotations

were in a single directory made publicly available under a requester-pays model, but we have now changed this to make the source code publicly available at no cost to the requester. It is not financially feasible for us to make large data sets available for download at no cost to the requester, as our research group suffered unsustainable costs of external users downloading large amounts of publicly available data from Broad Institute servers in late 2022; we believe that the requester-pays model is a good solution, as the cost to download a single copy of a data set is very modest, on the order of a few dollars per TB.)

Reviewer #4:

Kim and colleagues devised a robust statistical analysis method that leverages knowledge from GWAS to deduce disease associations specific to cell types in scRNA-seq or scATAC-seq datasets. They applied their approach to published scRNA-seq and scATAC-seq datasets, which span various developmental stages, brain regions, and sequencing batches. By integrating GWAS and single-cell profiling analyses, they identified cell types in the fetal or adult human brains that exhibit strong associations with neurological or psychiatric disorders. Moreover, they discovered that using their methods, scATAC-seq datasets are qualitatively more sensitive in identifying cell type-specific disease associations compared to scRNA-seq datasets, despite a general consistency in results between the two. Overall, their approach provides a way to bridge the gap between a multitude of GWAS targets and single-cell profiling datasets, which could enhance our understanding of disease mechanisms at the cellular level.

We thank the reviewer for providing an accurate summary of the manuscript.

In general, I agree with the comments from Reviewer #1 regarding the original manuscript. Reviewer #1 requested a more comprehensive explanation in the Methods section about how the authors derived their probability scores from scRNA-seq datasets, and also asked for the results of employing an alternative analytic strategy (Question #1). The reviewer also identified several errors in the manuscript related to the calculation of enrichment (Questions #2 and #5) and correlation (Question #3). Additionally, they suggested a method to enhance the statistical reliability of the analyses presented in the original manuscript (Question #4).

We thank the reviewer for providing an accurate summary of the comments from Reviewer #1 regarding the original manuscript.

The authors' response has adequately addressed all the queries and concerns raised by Reviewer #1.

We thank the reviewer for stating that all the queries and concerns raised by Reviewer #1 have been addressed in the revised manuscript.